# Association between sarcopenia level and metabolic syndrome

Su Hwan Kim[1,2], Ji Bong Jeong[1]*, Jinwoo Kang[1], Dong-Won Ahn[1], Ji Won Kim[1], Byeong Gwan Kim[1], Kook Lae Lee[1], Sohee Oh[3], Soon Ho Yoon[4], Sang Joon Park[4], Doo Hee Lee[5]

1 Department of Internal Medicine, Seoul National University College of Medicine, Seoul Metropolitan Government Seoul National University Boramae Medical Center, Seoul, Republic of Korea, 2 Health Care Center, Seoul Metropolitan Government Seoul National University Boramae Medical Center, Seoul, Republic of Korea, 3 Medical Research Collaborating Center, Seoul Metropolitan Government Seoul National University Boramae Medical Center, Seoul, Republic of Korea, 4 Department of Radiology, Seoul National University College of Medicine, Seoul National University Hospital, Seoul, Republic of Korea, 5 Department of Research and Development, MEDICALIP Co, Ltd., Seoul, Korea

* jibjeong@gmail.com

## Abstract

### Aims

Metabolic syndrome (MetS) increases the risk of diabetes mellitus (DM), cardiovascular disease (CVD), cancer, and mortality. Sarcopenia has been reported as a risk factor for MetS, non-alcoholic fatty liver disease, and CVD. To date, the association between sarcopenia and MetS has been investigated. However, there have been few studies on the dose-response relationship between sarcopenia and MetS. We investigated the association between sarcopenia and the prevalence of MetS. We also aimed to analyze the dose-response relationship between skeletal muscle mass and the prevalence of MetS.

### Methods

We enrolled 13,620 participants from October 2014 to December 2019. Skeletal muscle mass was measured using bioelectrical impedance analysis (BIA). Appendicular skeletal muscle mass (ASM) was divided by body weight (kg) and was expressed as a percentage (ASM x 100/Weight, ASM%). The quartiles of ASM% were calculated for each gender, with Q1 and Q4 being the lowest and highest quartiles of ASM%, respectively. The quartiles of ASM% were calculated for each gender, with Q1 and Q4 being the lowest and highest quartiles of ASM%, respectively. Linear regression and logistic regression analyses were used to compare the clinical parameters according to ASM%, adjusted for age, sex, obesity, hypertension (HT), DM, dyslipidemia (DL), smoking, alcohol intake, and C-reactive protein (CRP). Multiple logistic regression analysis was performed to determine the risk of MetS in each group.

### Results

A dose-response relationship was identified between ASM% and MetS. Sarcopenia was associated with an increased prevalence of MetS. After adjustment for age, sex, obesity,

**Data Availability Statement:** All relevant data are within the paper and its Supporting Information files.

**Funding:** MEDICALIP Co, Ltd. provided support in the form of salaries for author DHL, but did not

have any additional role in the study design, data collection and analysis, decision to publish, or preparation of the manuscript.

**Competing interests:** DHL is employed by MEDICALIP Co, Ltd. and was provided with support in the form of salaries. This does not alter our adherence to all PLOS ONE policies on sharing data and materials.

HT, DM, DL, smoking, alcohol intake, and CRP, sarcopenia remained significantly associated with MetS. For each 1 quartile increment in ASM%, the risk of MetS decreased by 56% (P< 0.001). After adjusting for age, sex, obesity, HT, DM, DL, smoking, alcohol intake, and CRP, the risk of MetS decreased by 25% per 1Q increment in ASM% (P < 0.001).

## Conclusions

Sarcopenia by BIA is independently associated with the risk of MetS and has a dose-response relationship.

## Introduction

Sarcopenia is defined as an age-related progressive loss of skeletal muscle mass [1,2]. With the global aging tendency of the world's population, sarcopenia has become a worldwide issue [3,4]. Loss of skeletal muscle mass has been reported as a risk factor for metabolic syndrome (MetS) [5–8], non-alcoholic fatty liver disease [9,10], carotid atherosclerosis and cardiovascular disease (CVD) [4,11,12]. In addition, sarcopenia causes arterial stiffness and hypertension (HT) [13]. Sarcopenia can limit physical and daily-life activities [14]. Sarcopenia also increases morbidity [15], disability [16], medical costs [17], and mortality [18].

MetS is a global health problem and is closely related to diabetes, with a prevalence of 34.75% in the US in 2012 [19]. MetS increases the risk of diabetes mellitus (DM), CVD [20,21], chronic liver disease and hepatocellular carcinoma [22], other cancers, and mortality [23–25]. Till date, the association between sarcopenia and MetS has been investigated [5–8]. However, there have been few studies on the dose-response relationship between sarcopenia and MetS.

In the current study, we investigated the association between sarcopenia and the prevalence of MetS. In addition, we aimed to analyze a dose-response relationship between skeletal muscle mass and the prevalence of MetS.

## Materials and methods

### Study population

We recorded 20,998 participants from October 2014 to December 2019, as they underwent a voluntary routine health checkup at the health care center of Seoul National University Boramae Medical Center. All data were fully anonymized before we accessed them. After excluding 2,627 participants with insufficient data and 4,621 participants who underwent repeated checkups, only the data from the first examination were included. Moreover, after excluding 130 participants with a history of malignancy, 13,620 participants were enrolled in our study (**Fig 1**). This study was approved by the Institutional Review Board of Boramae Medical Center (IRB No. 10-2020-234). The requirement for written informed consent was waived due to the retrospective nature of our study. Our study was conducted in accordance with the Helsinki Declaration.

### Data collection

The participants visited our health care center after an overnight 12-h fast. Clinical information and blood lab measurements were collected during the health checkup. Height and weight were measured when the subject was in a standing posture with a light examination gown and no shoes. Waist circumference (WC) was measured at the umbilicus level with the participants

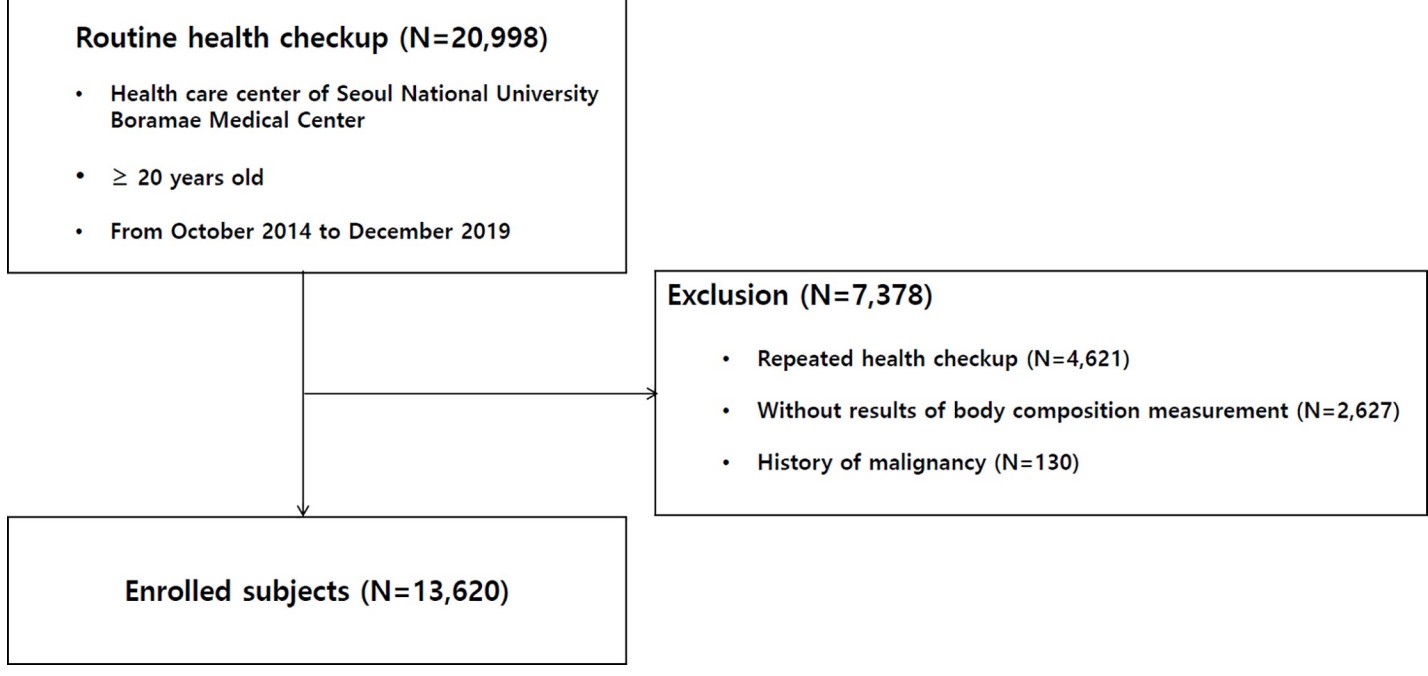

**Fig 1. Enrollment flow chart of patients.**

in a standing posture. Body composition analysis through bioelectrical impedance analysis (BIA) was performed with Inbody 720 (Biospace Co., Seoul, Korea) by a trained nurse following the manufacturer's protocol [26]. With Inbody 720, skeletal muscle mass and visceral fat area (VFA) were automatically calculated. Clinical information was collected: age, sex, systolic and diastolic blood pressure (BP), smoking, alcohol drinking habits, and medical history including HT and diabetes. Tests were performed to determine the following: total cholesterol, high-density lipoprotein cholesterol (HDL), low-density lipoprotein cholesterol (LDL), triglycerides (TG), glucose, aspartate aminotransferase (AST), alanine aminotransferase (ALT), uric acid, insulin level, and C-reactive protein (CRP).

### Definitions

BMI was defined as weight (kg) divided by height squared ($m^2$), and obesity was defined as BMI $\geq$ 25 kg/$m^2$ based on the criteria for the Asia-Pacific region. Underweight was defined as BMI < 18.5 kg/$m^2$ [27,28].

HT was defined as systolic BP $\geq$ 140 mmHg, diastolic BP $\geq$ 90 mmHg, or the use of antihypertensive medication. DM was defined as fasting plasma glucose $\geq$ 126 mg/dL, glycated hemoglobin level $\geq$ 6.5%, or the use of anti-diabetic medication including insulin.

MetS was defined when three or more of the following criteria was met: 1) WC male $\geq$ 102 cm, female $\geq$ 88 cm, 2) TG $\geq$ 150 mg/dL or the use of medication, 3) HDL male < 40 mg/dL, female < 50 mg/dL or the use of medication; 4) systolic BP $\geq$ 130 mmHg, diastolic BP $\geq$ 85 mmHg, or the use of antihypertensive medication, and 5) fasting plasma glucose $\geq$ 100 mg/dL or the use of anti-diabetic medication [29,30].

Homeostatic model assessment of insulin resistance (HOMA-IR) was calculated as [fasting glucose (mg/dL) $\times$ fasting insulin (µU/mL)]/405 [31].

Appendicular skeletal muscle mass (ASM) was calculated as the sum of the lean skeletal muscle mass of the bilateral upper and lower limbs. ASM was divided by body weight (kg) and

was expressed as a percentage (ASM x 100/Weight, ASM%). Sarcopenia was defined as ASM% < 29.0 in men and < 22.9 in women [32,33]. The quartiles of ASM% were calculated for each sex, with Q1 and Q4 being the lowest and highest quartiles of ASM%, respectively.

VFA was measured using Inbody 720 and was used to assess visceral obesity. Participants with VFAs ≥100 cm$^2$ were defined as the visceral obesity group [34–36].

## Comparison of Inbody 720 and computed tomography (CT) data

To evaluate the data of skeletal muscle mass measured by Inbody 720, we analyzed the correlation between BIA data and CT scans in participants who underwent body composition analysis using BIA and CT scans on the same day. Using CT, we measured VFA and total abdominal muscle area (TAMA) at the L3 vertebral level, which showed the highest correlation with visceral fat volume and whole body skeletal muscle in previous studies [37,38].

All abdominal CT scans were performed using a 64-slice multi-detector CT scanner (Brilliance 64 scanners; Philips Healthcare, Amsterdam, Netherlands). Pre-contrast CT images were analyzed using a commercially available segmentation software program (MEDIP Deep Catch v1.0.0.0, MEDICALIP Co. Ltd., Seoul, South Korea) to measure TAMA. After automatic segmentation, the reader selected the level of the inferior endplate of the L3 vertebra and extracted the TAMA at the corresponding level as previously described (**Fig 2**) [39]. The software contained 3D U-Net that was trained with 39,268 labeled CT images, providing an average dice similarity coefficient of 92.3% to 99.3% for muscle, abdominal visceral fat, and subcutaneous fat in the internal and external validation datasets. A clinically trained image analyst (DHL) reviewed and adjusted the results and finally a radiologist (SHY) confirmed the results.

## Statistical analysis

Continuous variables were expressed as mean ± standard deviation (SD). Categorical variables are presented as numbers and percentages. Linear regression and logistic regression analyses were used to compare the clinical parameters according to ASM%, adjusted for age, sex, obesity, HT, DM, DL, smoking, alcohol intake and CRP. Multiple logistic regression analysis was performed to determine the risk of MetS in each group. Crude odds ratios (ORs) were calculated with skeletal muscle mass at baseline. Model 1 was adjusted for age and sex. Model 2 was

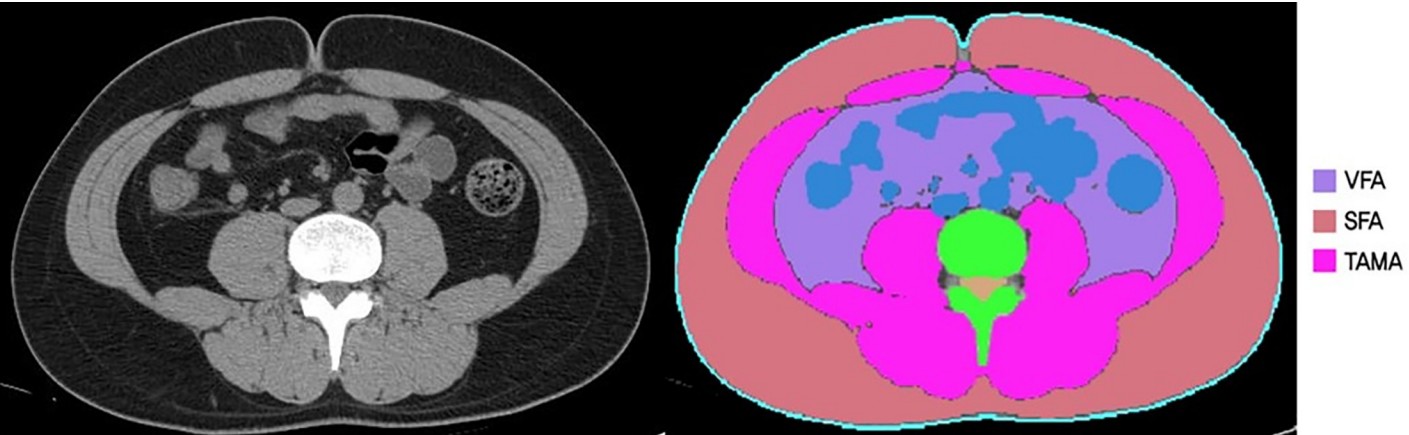

**Fig 2. Body morphometric evaluations of abdominal fat and muscle areas.** At the level of the inferior endplate of the L3 vertebra, a segmented axial computed tomography image showed the visceral fat area (VFA, cm$^2$), subcutaneous fat area (SFA, cm$^2$), and total abdominal muscle area (TAMA, cm$^2$), including all muscles on selected axial images (psoas, paraspinals, transversus abdominis, rectus abdominis, quadratus lumborum, and internal and external obliques).

adjusted for age, sex, and obesity. Model 3 was adjusted for age, sex, obesity, HT, DM, and DL. Model 4 was adjusted for age, sex, obesity, HT, DM, DL, smoking, and alcohol intake. Model 5 was adjusted for age, sex, obesity, HT, DM, DL, smoking, alcohol intake, and CRP. P-values less than 0.05 were considered statistically significant. All statistical analyses were conducted using IBM SPSS Statistics version 26 statistical software (IBM Corp., Armonk, NY).

## Results

### 1. Clinical characteristics according to ASM% quartiles

The mean age of the study population was 48.1 ± 13.1 years, and 54.5% was male. ASM% was 26.6 ± 2.9, 29.1 ± 2.5, 30.7 ± 2.5, 33.3 ± 2.9 years in the Q1, Q2, Q3, and Q4, respectively (P < 0.001, Table 1). As ASM% increased, the mean age and mean BMI decreased. From Q1 to Q4 of ASM%, WC, systolic and diastolic BP, VFA, LDL, TG, AST, ALT, fasting glucose, HbA1c, uric acid, insulin level, CRP, and HOMA-IR also significantly decreased (P < 0.001, Table 1). HDL increased in order from Q1 to Q4. As ASM% increased from Q1 to Q4, the proportions of HT, DM, obesity, and MetS decreased significantly (P < 0.001 in all, **Table 1**).

### 2. Association between ASM% and MetS

The prevalence of MetS was 34.2%, 16.8%, 11.3%, and 3.3% in Q1, Q2, Q3, and Q4 of ASM%, respectively (P for trend < 0.001, **Table 1**, **Fig 3**). Sarcopenia was associated with an increased prevalence of MetS (OR 5.306, 95% confidence interval [CI]; 4.656–6.046, P < 0.001). After adjustment for age, sex, obesity, HT, DM, DL, smoking, alcohol intake and CRP, sarcopenia remained significantly associated with MetS (OR 2.291, CI 1.874–2.801, P < 0.001, Model 5, **Table 2**).

In a stratified analysis according to visceral obesity, the association between sarcopenia and MetS was more prominent in participants without visceral obesity (OR 4.692 vs. OR 2.568, **Table 3**). In the stratified analysis according to obesity, the association between sarcopenia and MetS was more prominent in participants without obesity (OR 4.482 vs. OR 2.401, **Table 3**).

The prevalence of MetS according to sarcopenia was analyzed using age group stratification. In all age groups, the prevalence of MetS was significantly higher in the sarcopenia group (P < 0.05; **Fig 4**).

### 3. Association between sarcopenia and MetS with 4 or 5 criteria

Sarcopenia was associated with an increased prevalence of MetS with 4 or 5 criteria (OR 5.920, 95% CI; 4.974–7.045, P < 0.001). After adjustment for age, sex, obesity, HT, DM, DL, smoking, alcohol intake and CRP, sarcopenia remained significantly associated with MetS (OR 2.106, CI 1.681–2.639, P < 0.001, Model 5, **Table 4**) Sarcopenia was associated with an increased prevalence of severe MetS with 5 criteria (OR 10.453, 95% CI; 7.258–15.054, P < 0.001). After adjustment for age, sex, obesity, HT, DM, DL, smoking, alcohol intake and CRP, sarcopenia remained significantly associated with MetS (OR 3.073, CI 2.009–4.701, P < 0.001, Model 5, **Table 4**).

### 4. Quantitative association between sarcopenia and MetS

A dose-response relationship was identified between ASM% and MetS (**Fig 2**). The risk of MetS significantly decreased as ASM% increased, compared with Q1 (P < 0.001 in all, **Table 5**). For each 1 quartile increment in ASM%, the risk of MetS decreased by 56% (OR per 1Q increment 0.443, 95% CI; 0.422–0.466, P<0.001). The risk of MetS significantly decreased

**Table 1. Clinical characteristics according to ASM% quartiles.**

| Variables | Total | Q1 | Q2 | Q3 | Q4 | p for trend* |
|---|---|---|---|---|---|---|
| | N = 13620 | N = 3412 | N = 3392 | N = 3409 | N = 3407 | |
| Age (years) | 48.13±13.09 | 52.71±13.80 | 49.25±12.69 | 46.95±12.26 | 43.59±11.84 | <0.001 |
| Weight (kg) | 65.86±12.89 | 71.16±15.00 | 66.74±12.14 | 64.46±11.49 | 61.08±10.34 | <0.001 |
| Body mass index (BMI, kg/m²) | 23.71±3.42 | 26.64±3.59 | 24.25±2.56 | 22.89±2.38 | 21.04±2.22 | <0.001 |
| Waist circumference (cm) | 83.56±9.73 | 90.86±9.68 | 85.02±7.96 | 81.64±7.86 | 76.75±7.37 | <0.001 |
| Systolic blood pressure (mmHg) | 117.47±15.76 | 123.13±16.14 | 118.26±15.24 | 115.94±15.56 | 112.57±14.15 | <0.001 |
| Diastolic blood pressure (mmHg) | 79.10±10.98 | 81.76±11.35 | 79.86±10.77 | 78.53±10.82 | 76.25±10.19 | <0.001 |
| Visceral fat area (cm²) | 91.49±35.20 | 119.56±37.88 | 95.32±27.68 | 83.36±26.73 | 67.70±24.63 | <0.001 |
| ASM (kg) | 19.81±4.86 | 19.14±5.15 | 19.59±4.78 | 19.99±4.77 | 20.51±4.60 | <0.001 |
| ASM% | 29.94±3.63 | 26.64±2.85 | 29.06±2.53 | 30.71±2.46 | 33.33±2.87 | <0.001 |
| Cholesterol (mg/dL) | 196.36±36.21 | 200.40±39.13 | 198.88±36.36 | 196.30±35.40 | 189.86±32.76 | <0.001 |
| HDL (mg/dL) | 56.40±14.28 | 52.21±12.63 | 54.78±13.49 | 57.04±14.42 | 61.57±14.78 | <0.001 |
| LDL (mg/dL) | 118.20±33.61 | 122.38±36.06 | 120.98±34.05 | 118.19±32.82 | 111.32±30.20 | <0.001 |
| Triglyceride (mg/dL) | 110.40±76.84 | 131.84±86.41 | 117.59±84.46 | 106.92±74.15 | 85.25±48.68 | <0.001 |
| Glucose (mg/dL) | 94.44±19.96 | 100.28±23.93 | 95.45±20.38 | 92.95±17.30 | 89.09±15.49 | <0.001 |
| AST (IU/L) | 27.65±18.23 | 31.39±19.47 | 27.92±16.47 | 26.53±21.90 | 24.76±13.28 | <0.001 |
| ALT (IU/L) | 27.73±24.79 | 35.57±30.77 | 29.04±22.22 | 25.14±26.51 | 21.17±14.24 | <0.001 |
| Uric acid (mg/dL) | 5.25±1.34 | 5.52±1.43 | 5.28±1.30 | 5.16±1.31 | 5.02±1.25 | <0.001 |
| HbA1c (%) | 5.63±0.72 | 5.85±0.88 | 5.66±0.72 | 5.56±0.61 | 5.43±0.54 | <0.001 |
| Insulin | 9.65±5.78 | 12.44±7.84 | 9.74±4.40 | 7.80±2.91 | 6.80±2.78 | <0.001 |
| HOMA-IR | 2.42±1.65 | 3.25±2.13 | 2.46±1.31 | 1.83±0.73 | 1.61±1.13 | <0.001 |
| C-reactive protein (mg/dL) | 0.15±0.46 | 0.22±0.47 | 0.15±0.47 | 0.13±0.45 | 0.10±0.45 | <0.001 |
| Metabolic syndrome | 2238 (16.4) | 1165 (34.2) | 573 (16.8) | 386 (11.3) | 114 (3.3) | <0.001 |
| Hypertension | 4230 (31.1) | 1658 (48.7) | 1128 (33.1) | 899 (26.4) | 545 (16.0) | <0.001 |
| Diabetes mellitus | 1139 (8.4) | 498 (14.6) | 294 (8.6) | 226 (6.6) | 121 (3.6) | <0.001 |
| Obese status | | | | | | <0.001 |
| Obesity (BMI ≥ 25 kg/m²) | 4456 (32.7) | 2325 (68.3) | 1325 (38.9) | 676 (19.9) | 130 (3.8) | |
| Overweight (BMI 23–24.9 kg/m²) | 3266 (24.0) | 649 (19.1) | 1026 (30.1) | 1019 (29.9) | 572 (16.8) | |
| Normal (BMI 18.5–22.9 kg/m²) | 5345 (39.2) | 422 (12.4) | 1025 (30.1) | 1623 (47.7) | 2275 (66.8) | |
| Underweight (BMI < 18.5 kg/m²) | 553 (4.1) | 9 (0.3) | 29 (0.9) | 87 (2.6) | 428 (12.6) | |
| Smoking | 2381 (17.5) | 589 (17.3) | 584 (17.2) | 549 (16.1) | 659 (19.4) | 0.077 |
| Alcohol intake | 7224 (53.0) | 1660 (48.8) | 1822 (53.5) | 1827 (53.7) | 1915 (56.2) | <0.001 |

ASM, appendicular skeletal muscle mass; ASM%, ASMx100/Weight; HDL, high-density lipoprotein; LDL, low-density lipoprotein; AST, aspartate aminotransferase; ALT, alanine aminotransferase; HbA1c, glycated hemoglobin; HOMA-IR, Homeostatic Model Assessment of Insulin Resistance; Data are presented as mean± SD or number (%).

*From linear and logistic regression without any adjustment.

even in the Q2 group compared with the Q1 group (OR 0.389, 95% CI; 0.347–0.436, P < 0.001). After adjusting for age, sex, obesity, HT, DM, DL, smoking, alcohol intake and CRP, ASM% remained associated with the risk of MetS (Model 5, **Table 5**). In Model 5, the risk of MetS decreased by 25% per 1Q increment in ASM% (OR per 1Q increment 0.754, 95% CI 0.699–0.814, P < 0.001).

## 5. Correlation of skeletal muscle mass between Inbody 720 and CT

Among the population enrolled, CT scans were performed in 966 participants on the same day of Inbody 720. Thus, correlation analysis was conducted for these 966 participants. ASM

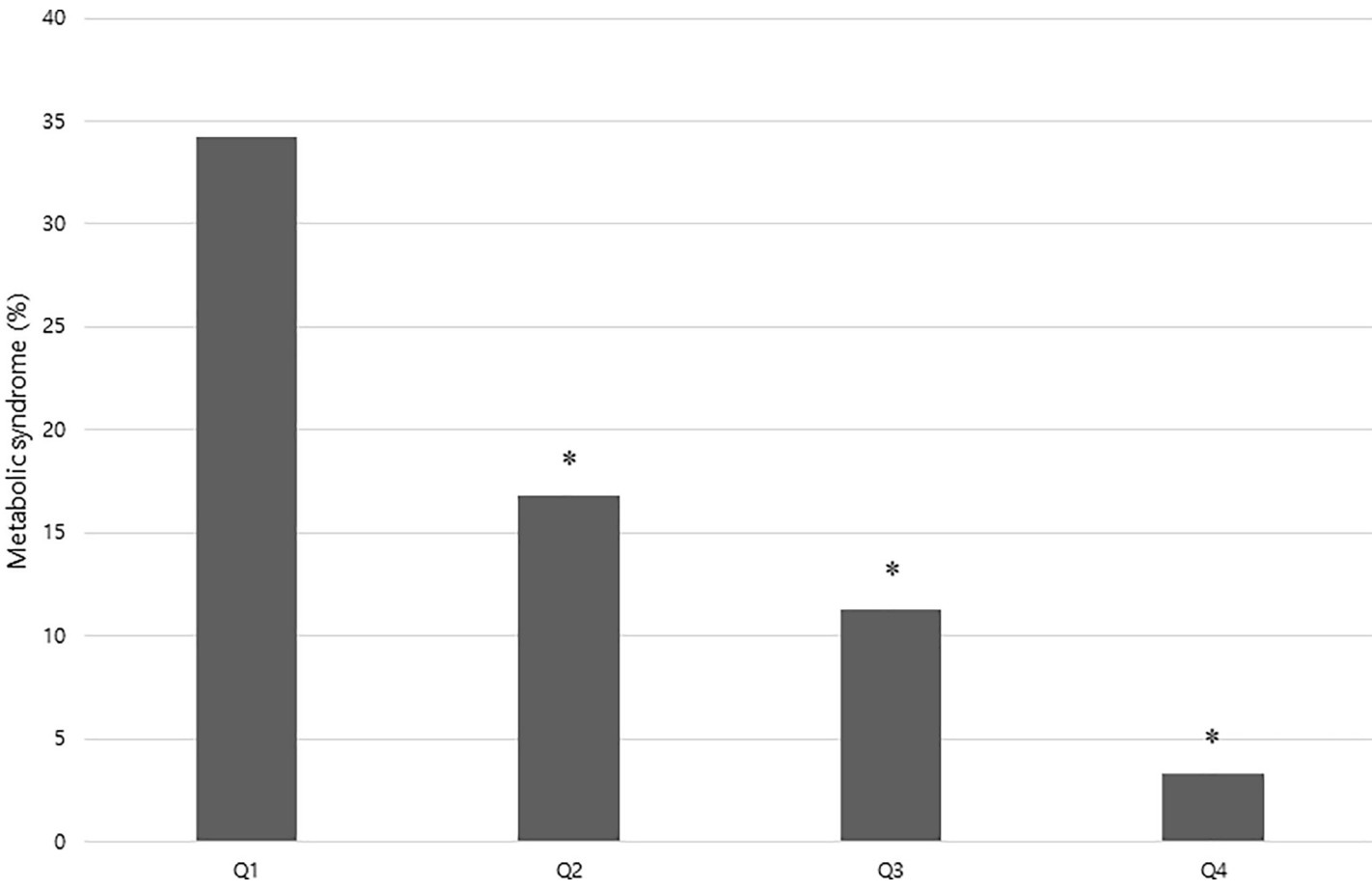

**Fig 3. Prevalence of metabolic syndrome according to ASM% (appendicular skeletal muscle mass x 100/Weight) quartiles.** *Significantly lower compared with the Q1 (P < 0.001).

**Table 2. Association between metabolic syndrome and sarcopenia.**

|         | Metabolic syndrome | | |
|---------|--------|-------------|---------|
|         | **OR** | **95% CI** | **p value** |
| Crude   | 5.306  | 4.656–6.046 | <0.001  |
| Model 1 | 4.414  | 3.847–5.065 | <0.001  |
| Model 2 | 2.254  | 1.950–2.605 | <0.001  |
| Model 3 | 2.325  | 1.903–2.840 | <0.001  |
| Model 4 | 2.328  | 1.905–2.844 | <0.001  |
| Model 5 | 2.291  | 1.874–2.801 | <0.001  |

Model 1: Adjusted for age, sex.

Model 2: Adjusted for age, sex, obesity.

Model 3: Adjusted for age, sex, obesity, hypertension, diabetes mellitus, dyslipidemia.

Model 4: Adjusted for age, sex, obesity, hypertension, diabetes mellitus, dyslipidemia, smoking, alcohol intake.

Model 5: Adjusted for age, sex, obesity, hypertension, diabetes mellitus, dyslipidemia, smoking, alcohol intake, CRP.

OR, odds ratio; CI, confidence interval; CRP, C-reactive protein.

Table 3. Stratified association between metabolic syndrome and sarcopenia.

| | Metabolic syndrome | | |
|---|---|---|---|
| | OR | 95% CI | p value |
| **Visceral obesity** | | | |
| Yes | 2.568 | 2.218–2.973 | <0.001 |
| No | 4.692 | 3.230–6.815 | <0.001 |
| **Obesity** | | | |
| Yes | 2.401 | 2.069–2.787 | <0.001 |
| No | 4.482 | 3.136–6.404 | <0.001 |
| **Underweight** | | | |
| Yes | 17.222 | 0.115–343.283 | 0.186 |
| No | 5.078 | 4.455–5.787 | <0.001 |
| **Sex** | | | |
| Male | 4.770 | 4.078–5.580 | <0.001 |
| female | 6.102 | 4.795–7.765 | <0.001 |

OR, odds ratio; CI, confidence interval; visceral obesity, VFAs $\geq$100 cm$^2$; obesity, BMI $\geq$ 25 kg/m$^2$; underweight, BMI < 18.5 kg/m$^2$.

measured by BIA was positively correlated with the TAMA measured by CT scan (R = 0.890, P < 0.001, **Fig 5**).

## Discussion

Our study showed that sarcopenia level measured by BIA was significantly associated with the risk of MetS in a dose-dependent manner. As ASM% increased from Q1 to Q4, the prevalence of MetS significantly decreased (**Table 1**, **Fig 2**). Not to mention the Q3 or Q4 groups, even individuals in the Q2 group had a significantly lower risk of MetS than those in the Q1 group (**Table 5**).

In the current study, sarcopenia was an independent risk factor for MetS regardless of age, sex, obesity, DM, HT, DL, smoking, alcohol intake and CRP levels (**Table 2**). The OR of MetS in participants with sarcopenia reached 2.266 after adjustment for age, sex, obesity, DM, HT, DL, smoking, alcohol intake and CRP levels. We adopted CRP as a variable as previous studies have shown the association between CRP and metabolic syndrome [40,41]. Our results are consistent with those of previous studies, which showed an association between sarcopenia and MetS [5–7,11,42]. We also analyzed the association between sarcopenia and more severe MetS with 4 or 5 criteria. The crude OR of severe MetS with 5 criteria in participants with sarcopenia was 10.453, which was higher than the 5.306 in the original MetS. After adjustment for age, sex, obesity, DM, HT, DL, smoking, alcohol intake and CRP levels, the OR of severe MetS with 5 criteria in subjects with sarcopenia was 3.119, which was higher than the 2.266 in the original MetS. We assume that severe MetS with 5 criteria may be more affected by skeletal muscle mass or sarcopenia. The strength of our study is that we demonstrated the dose-response relationship between sarcopenia and the risk of MetS. In our study, the risk of MetS significantly decreased for each 1 quartile increase of ASM%. Even after adjustment for age, sex, obesity, HT, DM, DL, smoking, alcohol intake and CRP, the risk of MetS significantly decreased by 25% per 1Q increase of ASM% (**Table 5**). The second strength of our study is that our study population included healthy individuals who voluntarily underwent routine health checkups. Thus, our results can be generalizable to the general healthy population. The large sample size is another strength of the current study.

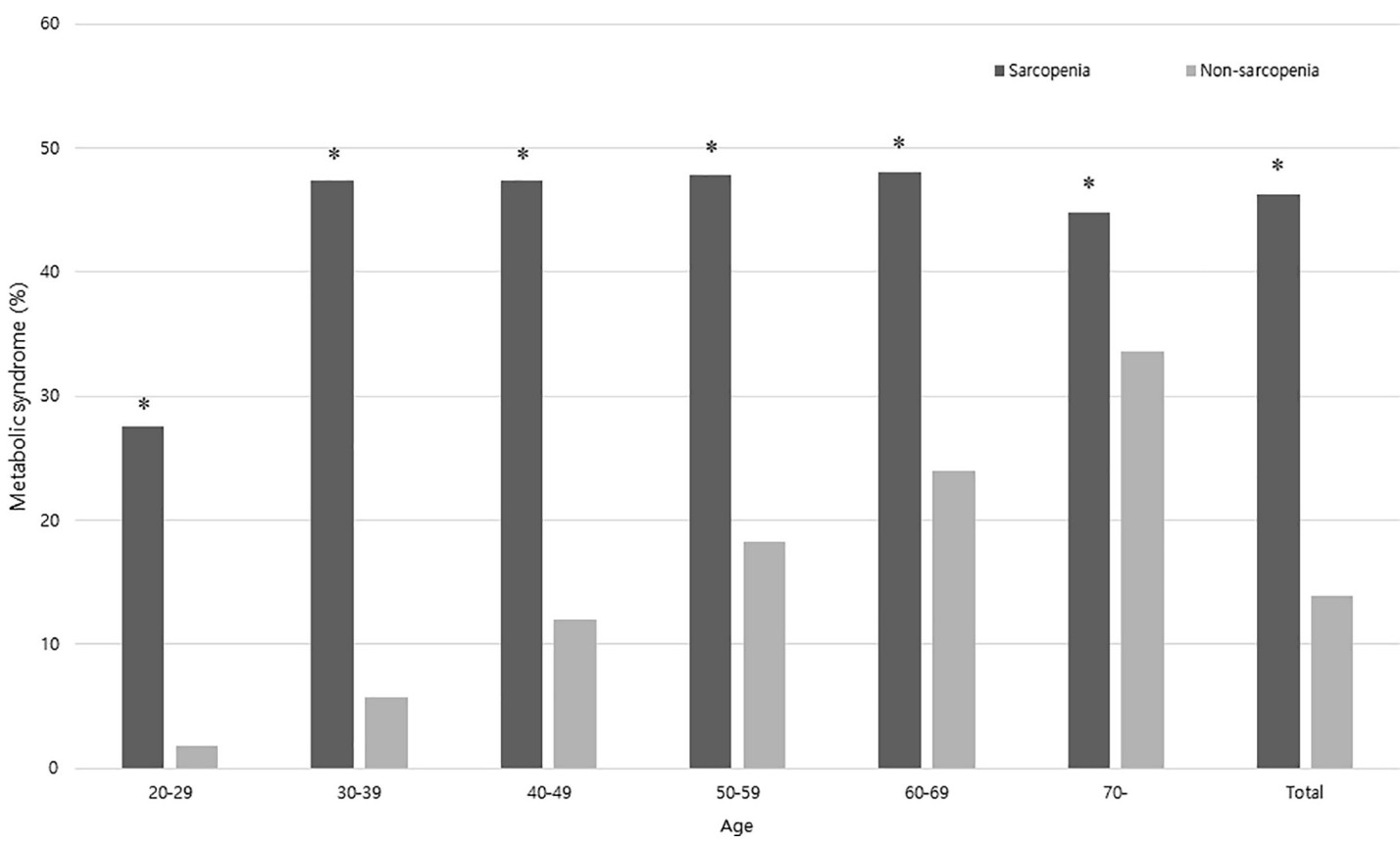

**Fig 4. The prevalence of metabolic syndrome in a 10-year age strata according to the presence of sarcopenia.** *Significantly higher compared with the non-sarcopenia group (P < 0.05).

We performed stratified analyses considering the possibility of other factors affecting the association between sarcopenia and MetS. In stratified analyses according to VFA, obesity, and sex, the association between sarcopenia and MetS was significant across all strata

**Table 4. Association between severe metabolic syndrome (4 or 5 criteria) and sarcopenia.**

| | Metabolic syndrome (4 or 5 criteria) | | | Metabolic syndrome (5 criteria) | | |
|---|---|---|---|---|---|---|
| | OR | 95% CI | p value | OR | 95% CI | p value |
| Crude | 5.920 | 4.974–7.045 | <0.001 | 10.453 | 7.258–15.054 | <0.001 |
| Model 1 | 5.073 | 4.222–6.096 | <0.001 | 10.285 | 6.921–15.286 | <0.001 |
| Model 2 | 2.375 | 1.960–2.878 | <0.001 | 3.605 | 2.442–5.323 | <0.001 |
| Model 3 | 2.195 | 1.756–2.745 | <0.001 | 3.243 | 2.130–4.940 | <0.001 |
| Model 4 | 2.182 | 1.744–2.730 | <0.001 | 3.170 | 2.078–4.835 | <0.001 |
| Model 5 | 2.106 | 1.681–2.639 | <0.001 | 3.073 | 2.009–4.701 | <0.001 |

Model 1: Adjusted for age, sex.

Model 2: Adjusted for age, sex, obesity.

Model 3: Adjusted for age, sex, obesity, hypertension, diabetes mellitus, dyslipidemia.

Model 4: Adjusted for age, sex, obesity, hypertension, diabetes mellitus, dyslipidemia, smoking, alcohol intake.

Model 5: Adjusted for age, sex, obesity, hypertension, diabetes mellitus, dyslipidemia, smoking, alcohol intake, CRP.

OR, odds ratio; CI, confidence interval; CRP, C-reactive protein.

**Table 5. Risk of metabolic syndrome in each quartile of sarcopenia.**

| | Metabolic syndrome | | |
| --- | --- | --- | --- |
| | OR | 95% CI | p value |
| Unadjusted | | | |
| Q1 | (Reference) | | |
| Q2 | 0.389 | 0.347–0.436 | <0.001 |
| Q3 | 0.246 | 0.216–0.279 | <0.001 |
| Q4 | 0.067 | 0.055–0.081 | <0.001 |
| Per 1Q | 0.443 | 0.422–0.466 | <0.001 |
| Model 1 | | | |
| Q1 | (Reference) | | |
| Q2 | 0.424 | 0.377–0.476 | <0.001 |
| Q3 | 0.286 | 0.251–0.326 | <0.001 |
| Q4 | 0.086 | 0.070–0.105 | <0.001 |
| Per 1Q | 0.481 | 0.457–0.506 | <0.001 |
| Model 2 | | | |
| Q1 | (Reference) | | |
| Q2 | 0.590 | 0.522–0.667 | <0.001 |
| Q3 | 0.514 | 0.446–0.593 | <0.001 |
| Q4 | 0.206 | 0.166–0.257 | <0.001 |
| Per 1Q | 0.645 | 0.609–0.684 | <0.001 |
| Model 3 | | | |
| Q1 | (Reference) | | |
| Q2 | 0.624 | 0.530–0.735 | <0.001 |
| Q3 | 0.607 | 0.504–0.732 | <0.001 |
| Q4 | 0.384 | 0.293–0.505 | <0.001 |
| Per 1Q | 0.751 | 0.696–0.810 | <0.001 |
| Model 4 | | | |
| Q1 | (Reference) | | |
| Q2 | 0.624 | 0.529–0.735 | <0.001 |
| Q3 | 0.608 | 0.504–0.733 | <0.001 |
| Q4 | 0.383 | 0.292–0.504 | <0.001 |
| Per 1Q | 0.751 | 0.696–0.810 | <0.001 |
| Model 5 | | | |
| Q1 | (Reference) | | |
| Q2 | 0.630 | 0.534–0.742 | <0.001 |
| Q3 | 0.615 | 0.510–0.742 | <0.001 |
| Q4 | 0.388 | 0.295–0.510 | <0.001 |
| Per 1Q | 0.754 | 0.699–0.814 | <0.001 |

Model 1: Adjusted for age, sex.

Model 2: Adjusted for age, sex, obesity.

Model 3: Adjusted for age, sex, obesity, hypertension, diabetes mellitus, dyslipidemia.

Model 4: Adjusted for age, sex, obesity, hypertension, diabetes mellitus, dyslipidemia, smoking, alcohol intake.

Model 5: Adjusted for age, sex, obesity, hypertension, diabetes mellitus, dyslipidemia, smoking, alcohol intake, CRP.

OR, odds ratio; CI, confidence interval; CRP, C-reactive protein.

(**Table 3**). The association between sarcopenia and MetS seemed more prominent in participants with low visceral fat or in non-obese participants. These findings are consistent with those of previous studies [7,43]. In a previous study by Moon *et al.*, sarcopenia was associated

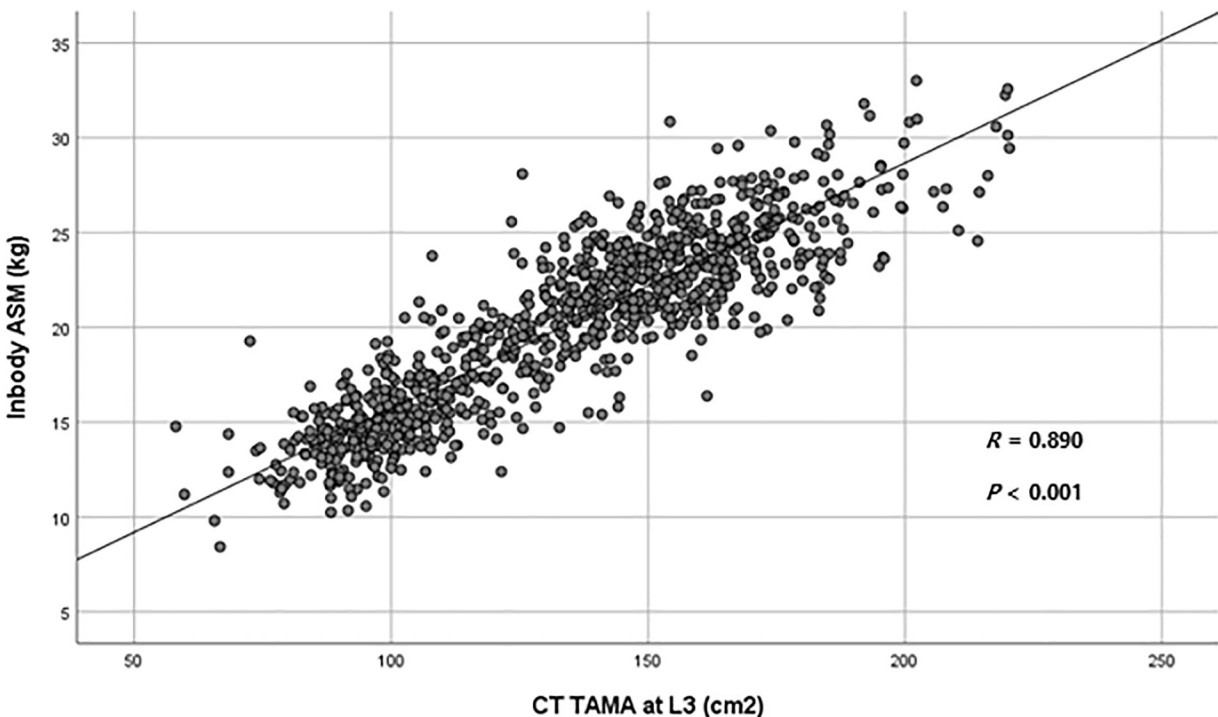

**Fig 5. Correlation between the appendicular skeletal muscle mass (ASM) measured by Inbody 720 and the total abdominal muscle area measured by computed tomography (CT) scan.** ASM, Appendicular skeletal muscle mass; TAMA, Total abdominal muscle area; CT, Computed tomography.

with insulin resistance, DM, and MetS in non-obese elderly subjects [7]. According to the results of previous and current studies, sarcopenia may be considered as a predictor of MetS susceptibility in the non-obese population. Considering the strong relationship between age and sarcopenia, the prevalence of MetS according to sarcopenia was analyzed using age group stratification (**Fig 4**). In all age groups, the prevalence of MetS was significantly higher in the sarcopenia group. This association between MetS risk and sarcopenia weakened as participants became older, though the association remained significant.

In the current study, skeletal muscle mass and VFA were measured using the BIA method (Inbody 720). The BIA method has strengths for use in clinical practice. Recently, BIA has been widely used with easy accessibility, quick assessment, safety, non-invasiveness, and cost-efficiency [27,44–46]. BIA has been reported to measure VFA and indicate the risk of MetS as precisely as CT [26,47]. In recent studies, BIA was used to assess skeletal muscle mass and to diagnose sarcopenia [48,49]. Our current study also showed that skeletal muscle mass measured by BIA was positively correlated with those calculated with CT scan. Based on the current study results and previous studies, BIA can be considered a valid option for measuring skeletal muscle mass in clinical practice. For measurement of skeletal muscle mass, CT, dual-energy X-ray absorptiometry (DEXA), and magnetic resonance imaging (MRI) may be other options. However, the use of CT and DEXA is limited due to the risk of radiation exposure, and MRI use is also limited because of cost [36].

Several mechanisms may affect the association between sarcopenia and MetS, including physical inactivity, insulin resistance, inflammation, and myokines [8,43]. Skeletal muscle is the main site of glucose uptake and utilization [50]. Thus sarcopenia is thought to increase insulin resistance and thereby induce DM and MetS [7]. However, in the current study, the

data of HOMA-IR results were available only in a small sample size (N = 305). Thus, we failed to analyze the association between sarcopenia and MetS adjusted for insulin resistance in the current study.

Our study has some limitations. First, this study was limited by its cross-sectional and single-centered retrospective design. It was difficult to assess the causal relationship between sarcopenia and MetS. Further prospective longitudinal cohort studies need to be conducted to validate whether sarcopenia is the cause of MetS. Second, our study population included healthy participants who underwent routine health checkups in a health care center. Thus, the results of our study are not generalizable to the diseased population or patients. Third, muscle strength was not evaluated in the current study. However, with only skeletal muscle mass measured by BIA, we could assess the risk of MetS easily, quickly, safely, and cost-efficiently. Fourth, exercise was not included in the variables and could not be evaluated in the analysis.

In conclusion, our study demonstrated that sarcopenia by BIA is independently associated with the risk of MetS and might have a dose-response relationship. Future studies that assess causal relationship between sarcopenia and MetS are needed using the data of subjects who underwent repeated health checkup. By measuring sarcopenia using BIA, the risk of MetS can be assessed easily, safely, and cost-efficiently. BIA can be used as an easy, useful, and important guide to identify participants with the risk of MetS.

## Supporting information

**S1 File. Data file.**
(XLSX)

## Author Contributions

**Conceptualization:** Ji Bong Jeong.

**Data curation:** Su Hwan Kim, Ji Bong Jeong, Sohee Oh, Soon Ho Yoon, Sang Joon Park, Doo Hee Lee.

**Methodology:** Su Hwan Kim, Ji Bong Jeong.

**Writing – original draft:** Su Hwan Kim.

**Writing – review & editing:** Su Hwan Kim, Ji Bong Jeong, Jinwoo Kang, Dong-Won Ahn, Ji Won Kim, Byeong Gwan Kim, Kook Lae Lee, Sohee Oh, Soon Ho Yoon, Sang Joon Park.

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
