## [Decision Letter · Decision Letter 0]

21 Jan 2021

PONE-D-20-39066

Association between sarcopenia level and metabolic syndrome

PLOS ONE

Dear Dr. Jeong,

Thank you for submitting your manuscript to PLOS ONE. After careful consideration, we feel that it has merit but does not fully meet PLOS ONE’s publication criteria as it currently stands. Therefore, we invite you to submit a revised version of the manuscript that addresses the points raised during the review process.

Major revisions are needed in the present form. 

See the Reviewers' comments carefully and respond them appropriately.

We look forward to receiving your revised manuscript.

Kind regards,

Masaki Mogi

Academic Editor

PLOS ONE

Journal Requirements:

2. In the ethics statement in the manuscript and in the online submission form, please provide additional information about the patient records/samples used in your retrospective study, including: a) whether all data were fully anonymized before you accessed them; b) the date range (month and year) during which patients' medical records/samples were accessed.

3.Thank you for stating the following in the Financial Disclosure section:

"The authors received no specific funding for this work."

We note that one or more of the authors are employed by a commercial company: MEDICALIP Co, Ltd.

Reviewers' comments:

Reviewer's Responses to Questions

**Comments to the Author**

1. Is the manuscript technically sound, and do the data support the conclusions?

Reviewer #1: Yes

Reviewer #2: Yes

2. Has the statistical analysis been performed appropriately and rigorously? 

Reviewer #1: Yes

Reviewer #2: Yes

3. Have the authors made all data underlying the findings in their manuscript fully available?

Reviewer #1: Yes

Reviewer #2: Yes

4. Is the manuscript presented in an intelligible fashion and written in standard English?

Reviewer #1: Yes

Reviewer #2: Yes

5. Review Comments to the Author

Reviewer #1: This article elucidated the relationship between the prevalence of metabolic syndrome (MetS) and the degree of muscle loss. I read very interestingly. I have some questions and propose some revisions to authors.

(1) I feel that Reference 1 and 2 are little bit old.

(2) You mentioned the relationship between sarcopenia and gallbladder polyps in Introduction part. Is this sentence necessary? I think the clinical importance of gallbladder polyps are not equal to other diseases, such as cancer, diabetes, which you mentioned.

(3) You cited Reference 28 as the definition of MetS. I think your definition is as same as the NCEP ATP-III, so you should add more suitable reference.

(4) In statistical analysis, I have three questions. First, why did you adopt C-reactive protein level as variable? Second, why did not you adopt smoking -/+ or alcohol intake -/+ as variables? I think the lean is also related to sarcopenia and Mets. Why did not you adopt body mass index as variables? In addition, I would like to know the percentage of obesity and lean participants in Table 1, if possible.

(5) The characteristics of patients with sarcopenia are heterogenous. Q1 group contained older, heavier, and less ASM and ASM (%) participants. Because you analyzed using many variables, such as age, sex, and obesity, I can accept your opinion. I would like to know results of subgroup analysis according to lean in Table 3.

Reviewer #2: 1. This study is a cross-sectional and single-centered retrospective research. Thus it was difficult to assess the causal relationship between sarcopenia and MetS. However, the researchers have the opportunity to study the causal relationship by the repeated health checkup(n=4621). I would like to be showed the corresponding results.

2. Some typos should be corrected, such as P13 TG≥ 150 g/dL

6. PLOS authors have the option to publish the peer review history of their article (what does this mean?). If published, this will include your full peer review and any attached files.

Reviewer #1: No

Reviewer #2: **Yes: **Zhi-hao Wang

---

## [Author Response · Author response to Decision Letter 0]

17 Feb 2021

Response Letter

Editors, PLOS ONE

Thank you for allowing the revision of our manuscript: ID PONE-D-20-39066 entitled “Association between sarcopenia level and metabolic syndrome” We revised our manuscript in accordance with the reviewers’ suggestions. Our responses to the comments are as follows.

Reviewer #1: 

This article elucidated the relationship between the prevalence of metabolic syndrome (MetS) and the degree of muscle loss. I read very interestingly. I have some questions and propose some revisions to authors.

Comment 1>

(1) I feel that Reference 1 and 2 are little bit old.

Reply: Thank you for your valuable comments. We added newer references.

Comment 2>

(2) You mentioned the relationship between sarcopenia and gallbladder polyps in Introduction part. Is this sentence necessary? I think the clinical importance of gallbladder polyps are not equal to other diseases, such as cancer, diabetes, which you mentioned.

Reply: Thank you for your valuable comments. Based on your comments, we eliminated gallbladder polyps from the sentence which you commented on. We revised our manuscript as follows.

MetS increases the risk of diabetes mellitus (DM), CVD, chronic liver disease and hepatocellular carcinoma, other cancers, and mortality. 

Comment 3>

(3) You cited Reference 28 as the definition of MetS. I think your definition is as same as the NCEP ATP-III, so you should add more suitable reference.

Reply: Thank you for your valuable comments. We added a suitable reference as you commented. 

Comment 4>

(4) In statistical analysis, I have three questions. First, why did you adopt C-reactive protein level as variable? Second, why did not you adopt smoking -/+ or alcohol intake -/+ as variables? I think the lean is also related to sarcopenia and Mets. Why did not you adopt body mass index as variables? In addition, I would like to know the percentage of obesity and lean participants in Table 1, if possible.

Reply: Thank you for your valuable comments. We revised our manuscript as follows with suitable references.

- We adopted CRP as a variable as previous studies have shown the association between CRP and metabolic syndrome. 

According to your comments, we added another model adjusted with smoking and alcohol intake. We revised Table 2, 4, 5. 

We intended to analyze our data from the aspect of MetS component. Thus, we adopted obesity as a binary variable instead of using BMI as a numerical variable. 

We added the percentage of obesity, overweight, normal BMI and lean (underweight) participants in Table 1. 

Comment 5>

(5) The characteristics of patients with sarcopenia are heterogenous. Q1 group contained older, heavier, and less ASM and ASM (%) participants. Because you analyzed using many variables, such as age, sex, and obesity, I can accept your opinion. I would like to know results of subgroup analysis according to lean in Table 3.

Reply: Thank you for your valuable comments. We performed subgroup analysis according to lean (underweight) in Table 3. We modified Table 3 according to your comments. 

Reviewer #2: 

Comment 1>

1. This study is a cross-sectional and single-centered retrospective research. Thus it was difficult to assess the causal relationship between sarcopenia and MetS. However, the researchers have the opportunity to study the causal relationship by the repeated health checkup(n=4621). I would like to be showed the corresponding results.

Reply: Thank you for your valuable comments. As you mentioned, it seems that causal relationship may be assessed by analyzing the data of subjects who underwent repeated health checkup. However, in the current study, we intended a cross-sectional study. We indeed have a plan to perform another study in the next step with the data of subjects who underwent repeated health checkup. As of now, it’s a pity that we can’t show you the data you want, but we’ll be able to show you when the next paper is completed. We sincerely ask for your kind understanding. 

Comment 2>

2. Some typos should be corrected, such as P13 TG≥ 150 g/dL

Reply: Thank you for your valuable comments. We revised our manuscript as follows.

TG ≥ 150 mg/dL or the use of medication

Thank you again for your insightful advice.

Yours sincerely,

Ji Bong Jeong, MD, PhD 

Associate Professor

Department of Internal Medicine

Seoul Metropolitan Government Seoul National University Boramae Medical Center

20 Boramae-ro 5-gil, Dongjak-gu

Seoul 07061, Republic of Korea

Phone: +82-2-870-2222

Fax: +82-2-870-3863

E-mail: jibjeong@gmail.com

---

## [Decision Letter · Decision Letter 1]

23 Feb 2021

PONE-D-20-39066R1

Association between sarcopenia level and metabolic syndrome

PLOS ONE

Dear Dr. Jeong,

Thank you for submitting your manuscript to PLOS ONE. After careful consideration, we feel that it has merit but does not fully meet PLOS ONE’s publication criteria as it currently stands. Therefore, we invite you to submit a revised version of the manuscript that addresses the points raised during the review process.

The Reviewer recommends some modification of conclusion before acceptance.

We look forward to receiving your revised manuscript.

Kind regards,

Masaki Mogi

Academic Editor

PLOS ONE

Journal Requirements:

Reviewers' comments:

Reviewer's Responses to Questions

**Comments to the Author**

1. If the authors have adequately addressed your comments raised in a previous round of review and you feel that this manuscript is now acceptable for publication, you may indicate that here to bypass the “Comments to the Author” section, enter your conflict of interest statement in the “Confidential to Editor” section, and submit your "Accept" recommendation.

Reviewer #1: All comments have been addressed

Reviewer #2: All comments have been addressed

2. Is the manuscript technically sound, and do the data support the conclusions?

Reviewer #1: Yes

Reviewer #2: Yes

3. Has the statistical analysis been performed appropriately and rigorously? 

Reviewer #1: Yes

Reviewer #2: Yes

4. Have the authors made all data underlying the findings in their manuscript fully available?

Reviewer #1: Yes

Reviewer #2: Yes

5. Is the manuscript presented in an intelligible fashion and written in standard English?

Reviewer #1: Yes

Reviewer #2: Yes

6. Review Comments to the Author

Reviewer #1: Thank you very much for answering my questions. I understood your reply well. I think this research article is very valuable. Thanks to authors and academic editor, I was able to have an important experience. As you plan to perform another study, I hope your next research will also be successful.

Comment 1＞

Thank you for your adding references.

Comment 2＞

After this, the topic of polyps was not mentioned, so I think the text became clear and concise. Thank you.

Comment 3＞

Thank you very much for adding a reference.

Comment 4＞

Thank you very much for telling me why you adopt CRP as a variable.

I have known the close relation between high-sensitivity CRP and chronic inflammation (as Ref 41 mentioned).

This time I understand the relation between CRP (not hs CRP) and metabolic syndrome (Ref40).

In addition, I appreciate your revising tables and adding the information of lean.

I understand your intention.

Comment 5＞

Thank you for conducting sub group analysis. I think the result of lean is also consistent with the sentence “the association between sarcopenia and MetS seemed more prominent in participants with low visceral fat or in non-obese participants”.

Reviewer #2: Still it was difficult to assess the causal relationship between sarcopenia and MetS. Therefore, the conclusion should be revised. If the author think that sarcopenia by BIA is independently associated with the risk of MetS and has a doseresponse relationship, the repeated health checkup would still be needed.I suggest modification of conclusion.

7. PLOS authors have the option to publish the peer review history of their article (what does this mean?). If published, this will include your full peer review and any attached files.

Reviewer #1: No

Reviewer #2: No

---

## [Author Response · Author response to Decision Letter 1]

24 Feb 2021

Response Letter

Editors, PLOS ONE

Thank you for allowing the revision of our manuscript: ID PONE-D-20-39066 entitled “Association between sarcopenia level and metabolic syndrome” We revised our manuscript in accordance with the reviewers’ suggestions. Our responses to the comments are as follows.

Reviewer #1: 

Thank you very much for answering my questions. I understood your reply well. I think this research article is very valuable. Thanks to authors and academic editor, I was able to have an important experience. As you plan to perform another study, I hope your next research will also be successful.

Comment 1>

(1) Thank you for your adding references.

Comment 2>

(2) After this, the topic of polyps was not mentioned, so I think the text became clear and concise. Thank you.

Comment 3>

(3) Thank you very much for adding a reference.

Comment 4>

(4) Thank you very much for telling me why you adopt CRP as a variable.

I have known the close relation between high-sensitivity CRP and chronic inflammation (as Ref 41 mentioned).

This time I understand the relation between CRP (not hs CRP) and metabolic syndrome (Ref40).

In addition, I appreciate your revising tables and adding the information of lean.

I understand your intention.

Comment 5>

(5) Thank you for conducting sub group analysis. I think the result of lean is also consistent with the sentence “the association between sarcopenia and MetS seemed more prominent in participants with low visceral fat or in non-obese participants”.

Reply: Thank you for your valuable comments. 

Reviewer #2: 

Comment 1>

1. Still it was difficult to assess the causal relationship between sarcopenia and MetS. Therefore, the conclusion should be revised. If the author think that sarcopenia by BIA is independently associated with the risk of MetS and has a dose-response relationship, the repeated health checkup would still be needed. I suggest modification of conclusion.

Reply: Thank you for your valuable comments. Based on your comments, we revised our manuscript as follows.

In conclusion, our study demonstrated that sarcopenia by BIA is independently associated with the risk of MetS and with a dose-response relationship. Future studies that assess causal relationship between sarcopenia and MetS are needed using the data of subjects who underwent repeated health checkup. By measuring sarcopenia using BIA, the risk of MetS can be assessed easily, safely, and cost-efficiently. BIA can be used as an easy, useful, and important guide to identify participants with the risk of MetS.

Thank you again for your insightful advice. 

Yours sincerely,

Ji Bong Jeong, MD, PhD 

Associate Professor

Department of Internal Medicine

Seoul Metropolitan Government Seoul National University Boramae Medical Center

20 Boramae-ro 5-gil, Dongjak-gu

Seoul 07061, Republic of Korea

Phone: +82-2-870-2222

Fax: +82-2-870-3863

E-mail: jibjeong@gmail.com

---

## [Decision Letter · Decision Letter 2]

8 Mar 2021

Association between sarcopenia level and metabolic syndrome

PONE-D-20-39066R2

Dear Dr. Jeong,

We’re pleased to inform you that your manuscript has been judged scientifically suitable for publication and will be formally accepted for publication once it meets all outstanding technical requirements.

Kind regards,

Masaki Mogi

Academic Editor

PLOS ONE

Additional Editor Comments (optional):

Accept with modification according to the Reviewer's comment.

See it and respond to the Reviewer's suggestion.

Reviewers' comments:

Reviewer's Responses to Questions

**Comments to the Author**

1. If the authors have adequately addressed your comments raised in a previous round of review and you feel that this manuscript is now acceptable for publication, you may indicate that here to bypass the “Comments to the Author” section, enter your conflict of interest statement in the “Confidential to Editor” section, and submit your "Accept" recommendation.

Reviewer #2: All comments have been addressed

2. Is the manuscript technically sound, and do the data support the conclusions?

Reviewer #2: Yes

3. Has the statistical analysis been performed appropriately and rigorously? 

Reviewer #2: Yes

4. Have the authors made all data underlying the findings in their manuscript fully available?

Reviewer #2: Yes

5. Is the manuscript presented in an intelligible fashion and written in standard English?

Reviewer #2: Yes

6. Review Comments to the Author

Reviewer #2: The conclusion should be modified. "our study demonstrated that sarcopenia by BIA is independently associated with the risk of MetS and might have a dose-response relationship"

7. PLOS authors have the option to publish the peer review history of their article (what does this mean?). If published, this will include your full peer review and any attached files.

Reviewer #2: No

---

## [Editor Report · Acceptance letter]

11 Mar 2021

PONE-D-20-39066R2 

Association between sarcopenia level and metabolic syndrome 

Dear Dr. Jeong:

I'm pleased to inform you that your manuscript has been deemed suitable for publication in PLOS ONE. Congratulations! Your manuscript is now with our production department. 

Kind regards, 

on behalf of

Dr. Masaki Mogi 

Academic Editor

PLOS ONE